# Consumer perceptions of antimicrobial use in animal husbandry: A scoping review

**Jaime R. Barrett[1]ᵒ, Gabriel K. Innes[2]ᵒ, Kelly A. Johnson[3]¤, Guillaume Lhermie[4,5,6], Renata Ivanek[7], Amelia Greiner Safi[8], David Lansing[1]***

**1** Department of Geography and Environmental Systems, University of Maryland Baltimore County, Baltimore, MD, United States of America, **2** Department of Biostatistics and Epidemiology, Rutgers School of Public Health, Piscataway, NJ, United States of America, **3** Flower-Sprecher Veterinary Library, College of Veterinary Medicine, Cornell University, Ithaca, NY, United States of America, **4** Department of Production Animal Health, University of Calgary, Calgary, Canada, **5** CIRAD, UMR ASTRE, Montpellier, France, **6** ASTRE, CIRAD, INRAE, Univ Montpellier, Montpellier, Université de Toulouse, Toulouse, France, **7** Department of Population Medicine and Diagnostic Sciences, College of Veterinary Medicine, Cornell University, Ithaca, NY, United States of America, **8** Department of Public and Planetary Health, College of Veterinary Medicine & Department of Communication, College of Agriculture and Life Sciences Cornell University, Ithaca, NY, United States of America

ᵒ These authors contributed equally to this work.
¤ Current address: Charles Allen Cary Veterinary Medical Library, College of Veterinary Medicine, Auburn University, Auburn, AL, United States of America
* dlansing@umbc.edu

**Data Availability Statement:** The charting data that is the basis of our analysis can be found at: https://osf.io/b5cdq/.

## Abstract

Antimicrobial use in animal agriculture is often perceived to play a role in the emerging threat of antimicrobial resistance. Increased consumer awareness of this issue places pressure on animal husbandry to adopt policies to reduce or eliminate antimicrobial use. We use a scoping review methodology to assess research on consumer perceptions of antimicrobial drugs in meat products in the United States, Canada, or the European Union. Evaluating peer-reviewed and grey literature, we included studies for assessment if they met these topical and geographic requirements, involved primary data collection, and were originally published in English. Our screening process identified 124 relevant studies. Three reviewers jointly developed a data charting form and independently charted the contents of the studies. Of the 105 studies that measured consumer concern, 77.1% found that consumers were concerned about antimicrobial use in meat production. A minority of studies (29.8% of all studies) queried why consumers hold these views. These studies found human health and animal welfare were the main reasons for concern. Antimicrobial resistance rarely registered as an explicit reason for concern. A smaller group of studies (23.3%) measured the personal characteristics of consumers that expressed concern about antimicrobials. Among these studies, the most common and consistent features of these consumers were gender, age, income, and education. Regarding the methodology used, studies tended to be dominated by either willingness-to-pay studies or Likert scale questionnaires (73.64% of all studies). We recommend consideration of qualitative research into consumer views on this topic, which may provide new perspectives that explain consumer decision-making and mentality that are lacking in the literature. In addition, more research into the difference

**Funding:** Financial Disclosure: JRB, GKI, GL, RI, AGS, DL received funds from the United States Department of Agriculture, National Institute for Food and Agriculture grant # 2019-67017-29114. JRB, DL received funds from United States Department of Agriculture, National Institute for Food and Agriculture grant #2018-68003-27467 The funders had no role in the study design, data collection and analysis, decision to publish, or preparation of the manuscript.

**Competing interests:** The authors have declared that no competing interests exist.

between what consumers claim is of concern and their ultimate purchasing decisions would be especially valuable.

## Introduction

The rise of antimicrobial-resistant organisms threatens human and animal health [1]. In livestock production systems, antimicrobials have been used for prevention and treatment of disease and, in many countries, growth promotion [2, 3]. Antimicrobial use in animal husbandry has been linked to antimicrobial-resistant bacterial infections in humans [4]. To address public concern about antimicrobial resistance, regulation has been promulgated to limit the use of certain drugs in animal husbandry [5]. A recent amendment in 2017 to the Veterinary Feed Directive of the United States Department of Agriculture's (USDA) Animal Drug Availability Act of 1996 changed drug use allowances in U.S. animal agriculture industries. This amendment prohibits the use of medically important antimicrobials in food-producing animals for growth promotion or to improve feed efficiency, and requires approval from the overseeing veterinarian for antimicrobials that are administered via feed and water [6]. In addition to this federal regulation, state governments such as California [7] and Maryland [8] have implemented laws in 2018 that also restrict antimicrobial use in animal husbandry. As with the Veterinary Feed Directive, the effectiveness of these bills has yet to be assessed.

Governmental regulatory efforts may prove to be an important step in decreasing antimicrobial resistance development in animal husbandry. However, private industry standards are increasingly the impetus for change in the agri-food system [9]. Many agricultural standards are voluntary and put forth by private companies and trade associations (e.g., national dairy associations) to avoid further government [9, 10]. These shifts are also driven by the need to maintain their consumer base in a saturated market and therefore attempt to address consumer demand for safe food of a uniform quality that is produced under conditions consumers can support [9, 10]. For example, large animal product purchasers, such as McDonalds and public-school systems, have committed to using "antibiotic free" animal products [11, 12]. Consumer attitudes may reflect confusion about modern production practices. For example, some consumers purchase "raised without antibiotics" animal products because of their concerns for animal welfare [13, 14]. However, these consumers may not understand that antimicrobials are necessary for the prevention and treatment of diseases in animals, and thus a complete ban could lead to increased animal suffering if they are withheld in cases of clinical infections [2, 14].

Despite potential consumer confusions about the role of antimicrobials in animal agriculture, such perceptions are important drivers of animal husbandry practices across the wider commodity chain. The adoption of market products with labels such as "no antibiotics ever" in the poultry industry, for example, exerts downward pressure on the production practices of broiler integrators [15]. Similar consumer-driven pressures have been noted across other animal production industries as well [16]. In short, the increasing prevalence of "antibiotic free" labels on food, and emerging evidence that consumers will pay more for meat with this label, mean that consumers may influence the governance of wider food systems.

While research on consumer preferences for meat purchase and consumption is explored in the scientific literature, the salience of antimicrobial use in animal-based food production calls for a closer examination of the scientific evidence on this topic. To the best of our knowledge, no review has investigated consumer perceptions of antimicrobial use in animal

husbandry; we fill this gap with a scoping review. Due to the similar regulatory infrastructure and levels of economic development across these countries, we conducted a review of research on consumer perceptions in the United States, Canada, and the European Union. Within this geographically limited focus, we aim to summarize the extant research on this topic, identify research areas that are both well-studied and ignored, and understand what consumers see as the risks and benefits of antimicrobial use in animal husbandry. Further, we identify the methods used to assess consumer perception in order to gauge existing methodological gaps in the literature.

## Materials and methods

This review was completed in compliance with the guidelines outlined in the Preferred Reporting Items for Systematic Reviews and Meta-Analyses Extension for Scoping Reviews (PRISMA-ScR) [16]. PRISMA Sc-R represents a checklist of essential and optional reporting items that maximizes a scoping review's methodological and reporting quality by increasing transparency, comprehensiveness, and reproducibility while minimizing bias. The review team was composed of experts in the field (RI, AGS, GL, DL), a research librarian (KAJ), and doctoral students (GKI, JRB).

### Research question and definitions

This review aims to identify and describe peer-reviewed and grey literature relevant to the research question: "What are consumer perceptions concerning antimicrobial use in animal husbandry in the United States, Canada, and the European Union?" and utilizes the following definitions.

**Consumer perceptions and attitudes.** We define consumers as individuals who purchase food. Of particular interest to this review are consumers who purchase animal-based products for personal or familial consumption or consumers who choose not to purchase animal-based products, and their reasoning. Perception encompasses awareness, understanding and interpretation of an individual's surroundings. Attitude includes, but is not limited to, one's thoughts, feelings, beliefs, and willingness to pay for food. In combination this review will assess the level of awareness and understanding of general audiences in regard to antimicrobials in animal products and animal agriculture.

**Antimicrobials.** For the purposes of this review, we define antimicrobials as drugs that are administered to patients to treat and/or prevent infection, illness, and/or other health problems resulting from exposure to microbial organisms. These can include antibiotics, antifungals, antiprotozoals, and antivirals. For the purposes of this review we are interested in antimicrobials administered to maintain the health and well-being of agricultural animals raised for human consumption, of which antibiotics (i.e., drugs that target bacteria) are primarily used.

**Animal agriculture.** For this review, we define animal agriculture as the husbandry of animals for consumption of their meat or other products. Animals included in this category are as follows: ruminants (cattle, sheep, goats, bison), pigs, poultry (chickens, turkeys, ducks), and fish (shellfish and finfish).

A protocol for this review was registered on the Open Science Framework (osf.io) on August 8, 2019, and can be located at https://osf.io/rp9ak/. An amendment was made at the initiation of full text screening and was uploaded on December 23, 2020, and can be located at https://osf.io/mcd93/.

### Search strategy, databases, and grey literature sources

A comprehensive search was developed for CAB Abstracts and Global Health (CABI) using search terms related to consumer perceptions, antimicrobials, and animal agriculture.

The search was translated and run in ABI/Inform (ProQuest), AGRICOLA (EBSCOhost), BIOSIS Citation Index (Clarivate Analytics), Business Source Complete (EBSCOhost), FSTA/Food Science and Technology Abstracts (Clarivate Analytics), Medline (PubMed), ProQuest Dissertations and Theses Global (ProQuest), VetMed Resource (CABI), and Web of Science Core Collection (Clarivate Analytics). Database searches were executed on August 14, 2019, without date or language restrictions, and updated on May 10, 2021 with no language restriction but a date restriction of August 2019 forward. Grey literature sources were searched between August 24, 2019 and September 24, 2019, and again from May 19–28, 2021 Publications and factsheets were manually searched in: Agriculture and Agri-Food Canada; Canadian Antimicrobial Resistance Surveillance System; Centers for Disease Control and Prevention (CDC) Antibiotic/Antimicrobial Resistance Reports and Publications; Environmental Working Group; European Commission; European Food Safety Authority; Food and Agriculture Organization (FAO) of the United Nations; FDA Antimicrobial Resistance Information; FDA Guidance Documents; Pew Charitable Trusts Antibiotic Resistance Project; USDA Economic Research Service; and World Health Organization (WHO). Search terms, databases, and number of results for each of the database searches are available at https://osf.io/p82fg/. Search terms, sources, and number of results for the grey literature searches are available at https://osf.io/frxsw/.

### Citation management

References returned from all database and grey literature searches were imported or manually entered into Zotero citation management software (Version 5.0.73). Following deduplication in Zotero, the remaining records were imported to the screening software Covidence (covidence.org), where additional duplicates were identified. The remaining records were eligible for inclusion in the review.

### Study selection and screening

Studies were considered eligible for inclusion in this review if they: (1) include reference to antimicrobial use in food animals, (2) include consumer viewpoints about antimicrobial use in food animals, (3) describe studies about consumer populations in the United States, Canada, or the European Union, (4) are originally published in English, and (5) describe primary data collection. Studies were excluded if they did not satisfy all inclusion criteria.

Each record was evaluated against the predetermined inclusion criteria by two independent reviewers at the level of title and abstract. Those records that were not eliminated at this stage were then considered by two independent reviewers at the full-text level. For both the title and abstract stage and full-text stage, conflicts were resolved either by consensus or by a third, independent reviewer.

Number of sources included at each stage of retrieval, screening, and data extraction, as well as reasons for exclusion at the full-text screening phase, are indicated in the PRISMA diagram (Fig 1). As prescribed for scoping reviews [16, 17], risk of source bias was not evaluated during consideration for inclusion.

### Data charting and analysis

Based on trends and concepts identified during screening, a list of relevant data categories was developed to guide data extraction. Each of the three main reviewers (GKI, JRB, DL) extracted data from five papers to evaluate the list's comprehensiveness. Additional categories were added after this pre-testing, as well as during the extraction process when new trends were identified. One of the three main reviewers extracted data from each of the studies. Multiple

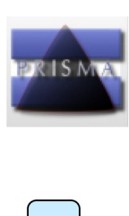

## PRISMA 2009 Flow Diagram

**Identification**

Records identified through database searching
(n = 6383)

Additional records identified through other sources
(n = 199)

Records after duplicates removed
(n = 3560)

**Screening**

Records screened
(n = 3560)

Records excluded
(n = 3192)

**Eligibility**

Full-text articles assessed for eligibility
(n = 367)

Full-text articles excluded, with reasons
(n = 243)
- No primary data (149)
- Not retrievable, insufficient or duplicate data (41)
- Not about consumer perception (22)
- Not about antimicrobials in animal agriculture (14)
- Not in English (10)
- Non-US/Canada/EU (8)

**Included**

Studies included in qualitative synthesis
(n = 124)

**Fig 1. PRISMA flow chart.** Number of sources found at each stage of retrieval, screening, and data extraction.

discussions throughout this process were used to ensure consistency. The data from this charting process is available at: https://osf.io/b5cdq/. This data includes charting from both the initial and the updated searches.

Extracted data include: study type (qualitative or quantitative), publication source, author affiliation, publication date, country of study population, number of participants, response

rate, population selection criteria, product of study, data collection method, qualitative and quantitative models and associated analysis units (willingness-to-pay and Likert scale), specific results about perceptions of antimicrobial use and several binary variables for statistical analysis. The extracted data were coded in anticipation of statistical analysis.

**Exclusion criteria.** When developing the protocol for this review, we limited our scope to studies that investigated consumers in the U.S., Canada, and the European Union (including the United Kingdom). We made this decision for a number of reasons. First, these countries have similar regulatory environments, with state agencies that make science-based decisions about regulations of drug use and food safety. Second, these countries have well-developed trade associations with each other. Third, these countries are at a relatively similar level of economic development, meaning that the type of food consumer in these places is roughly comparable, and more so than if we had expanded the geographical scope of our search to include more lower- and middle-developed countries. Finally, we had the language capacity to include English only texts. Therefore, some otherwise relevant Canadian and European studies were excluded. Between the title and abstract stage and full text screening stage of this review, we further decided to exclude any texts that did not contain primary data collection (reflected in the amended protocol). As a result, most of the news articles and opinion pieces that were originally included were excluded. This decision was made in an effort to avoid bias given that we could not ensure that all non-academic texts about this topic were captured. However, several news articles with extractable data were included in the final analysis because they cited studies that were not otherwise captured through database and grey literature searches. Although our search strategy was comprehensive in its use of "antimicrobial" and the other associated terms listed above, extracted studies about consumer concern all focused on antibiotic use as opposed to antimicrobial use; and the term "antibiotic" was overwhelmingly used in these studies. For this reason, we use the more specific term "antibiotics" for the results and discussion sections.

**Analysis of consumer concern.** To answer our proposed question, we performed additional analysis on the studies that measured consumer concern in an attempt to summarize them based on information available in the studies. We had three categories of concern: concerned, mixed concern, and not concerned. For manuscripts that utilized Likert scale surveys, studies were classified as finding that consumers were "concerned" if there was, on average, a higher than neutral level of agreement with a statement that expressed concern about antibiotic use. Conversely, Likert surveys that indicated a lower than neutral level of agreement for similar statements were coded as finding that consumers were "not concerned." Willingness-to-pay studies that showed consumers were willing to pay more for food with antibiotic-free traits (at a statistically significant level) were labeled as studies that showed consumers are "concerned." Similarly, willingness-to-pay studies that *failed* to find consumers would pay more for antibiotic-free food were coded as having found consumers to be "not concerned." Some studies found that consumers agreed with some concern-type statements while disagreeing with others; such studies were labeled as "mixed concern." Qualitative studies were read for the authors' main conclusions about consumer's perceptions, and were then appropriately coded.

Reasons for consumer concern were identified, and each reason was given a unique identifier for analysis. For studies that investigated the characteristics of people who are concerned about antimicrobial use, statistically significant demographics (e.g., gender, religion) were tallied. Most studies that evaluated consumer characteristics concluded that multiple characteristics were associated with antibiotic use concerns.

The coded spreadsheet of extracted data was imported into Stata (Version MP 16) to perform descriptive statistical analysis. Statistical tables including frequencies and percentages

were generated to identify dominant categories for each extracted data type. More in-depth analysis of results was used in conjunction with frequency and percentage statistics to assess for gaps in the research.

## Results

Study selection and exclusion criteria are summarized by the PRISMA flow diagram illustrated in Fig 1. From the 3,560 citations imported for title and abstract screening, 368 were chosen for full text screening and 124 met inclusion criteria. Table 1 shows publication date ranges, study locations, and author affiliations for studies ultimately selected for inclusion. Due to the

**Table 1. Timeline and source characteristics from the extracted texts.**

| Study Characteristics | No. | % |
|---|---|---|
| *Publication Date* | | |
| Pre-2009 | 38 | 30.6% |
| 2010–2015 | 34 | 27.4% |
| 2016–2020 | 52 | 42.9% |
| *Publication Type* | | |
| Academic Journal | 83 | 66.9% |
| Book | 1 | 0.8% |
| Dissertation | 7 | 5.6% |
| Thesis | 5 | 4.0% |
| News Article | 9 | 7.3% |
| White Paper | 2 | 1.6% |
| Report | 7 | 5.6% |
| Trade Journal | 4 | 3.2% |
| Conference/Workshop Paper | 3 | 2.4% |
| Website | 1 | 0.8% |
| Datasheet | 2 | 1.6% |
| *Author Affiliation* | | |
| University | 89 | 71.8% |
| Government | 7 | 5.6% |
| Experiment Station | 2 | 1.6% |
| Industry | 8 | 6.5% |
| Think Tank | 2 | 1.6% |
| Advocacy Group | 1 | 0.8% |
| University and Government | 1 | 0.8% |
| University and Industry | 1 | 0.8% |
| Government and Industry | 2 | 1.6% |
| Group/Association | 4 | 3.2% |
| Unspecified | 7 | 5.6% |
| *Country of Study* | | |
| United States | 67 | 54.0% |
| Canada | 12 | 9.7% |
| Germany | 7 | 5.6% |
| Single European Union Country | 17 | 13.7% |
| United States and Canada | 3 | 2.4% |
| Multiple European Union Countries | 10 | 8.1% |
| Mixed European and North American Countries | 5 | 4.0% |
| Unspecified | 3 | 2.4% |

inclusion criterion of primary data collection, most of the relevant texts were published in academic journals (66.9%) with news articles a distant second (7.3%); the remaining 25.8% were a mix of other publication types, such as dissertations. Publications before 2009 comprise 30.6% of the sample, 27.4% were published between 2010 and 2015, and 41.9% were published between 2016 and 2021. The majority of research was conducted in the U.S. (54.0%). Canada (9.7%) and Germany (5.6%) were the next most commonly studied countries. The goal of this review was to compare studies that exist among populations in similar regulatory environments and levels of economic development. This desire for similarity was the basis of our decision to restrict our searches to the U.S., Canada, and the EU. It was not our intention to compare differences across these different geographic sites. Finally, most studies (71.8%) were conducted solely by university researchers. Government researchers accounted for 5.6% of studies, industry researchers comprise another 6.5%, and 5.6% of papers did not specify their affiliation. The remaining 10.4% of papers were a mix of think tanks, advocacy groups, experiment stations and various collaborations between industry, academia, and government (see Table 1 for a more specific breakdown).

Many animal agriculture products were investigated, with no single type dominating the body of literature (Table 2). The most frequently investigated single product categories are pork (15.3%) and beef (12.9%), poultry (10.5%), and dairy (10.5%). The most frequent product category is the generic category (24.2%), which includes studies that investigated "food," "organic food," "meats," and/or other similarly broad categories. Multiple product studies were tied for the second most frequent category (15.3%) and included a range of product combinations from pork and eggs to dairy and apples.

Studies often had multiple themes but those tallied in Table 2 were identified by reviewers as the primary focus of each study. We found 18 distinct research themes for which antibiotic perception data could be assessed. Few publications (12.9%) had a central focus on consumer perceptions of antibiotics. More commonly, antibiotics were one of several consumer concerns that were measured in a study. Of the studies with a main focus on antibiotic use, dairy (n = 6) and beef (n = 4) were the most common, followed by pork (n = 2). Other core topics for studies include production characteristics (23.2%), food safety (16.1%), and credence claims/product attributes (10.5%). The production characteristics category includes any publication that focuses on agricultural practices and other aspects of production, e.g., rearing practices, conventional versus organic production, and other similar foci. The credence claims/product attributes category encompasses publications with a primary focus on perceptions of particular food characteristics, e.g., raised without antibiotics, natural, organic, and other labeled product attributes.

The publications under review were dominated by quantitative methods (82.3%; Table 3). Qualitative methods—including interviews, focus groups, and document analysis—were used in 11.3% of the studies, and mixed quantitative/qualitative techniques were used in 6.5% of studies. Data collection was divided into five categories: surveys (56.5%), choice experiments (6.5%), qualitative methods (6.5%), document and literature analysis (6.5%), and mixed approaches (21.0%). Four studies (3.2%) did not identify their method of data collection. In terms of specific quantitative methodologies, willingness-to-pay studies (33.9%) and Likert scale surveys (39.5%) were the most utilized techniques to ascertain consumer perceptions.

Economics is the dominant field of research that investigated consumer attitudes and concerns with antibiotic use in animal agriculture with 44.8% of the texts describing an economic or marketing component of consumer perceptions. Of these papers, 17.9% did not collect original data and 12.5% had unclear or missing information. The remaining publications (69.6%) consisted of consumer surveys administered to a varying number of people (min: 154, max: 7795). These studies used a variety of econometric analyses; 14 studies used a choice

**Table 2. Product and theme focus of extracted texts.**

| Study Characteristics | No. | % |
|---|---|---|
| *Product* | | |
| Beef | 16 | 12.9% |
| Pork | 19 | 15.3% |
| Poultry | 13 | 10.5% |
| Dairy | 13 | 10.5% |
| Seafood | 6 | 4.8% |
| Other Single Products | 1 | 0.8% |
| Mixed Products | 19 | 15.3% |
| Generic Categories | 30 | 24.2% |
| Unspecified | 7 | 5.6% |
| *Themes* | | |
| Antibiotic Use | 16 | 12.9% |
| Production Characteristics | 28 | 22.6% |
| Food Safety | 20 | 16.1% |
| Credence Attributes | 13 | 10.5% |
| Organic | 8 | 6.5% |
| Labels | 8 | 6.5% |
| Food Quality | 6 | 4.8% |
| Animal Welfare | 6 | 4.8% |
| Risk | 5 | 4.0% |
| Natural | 3 | 2.4% |
| Environmental Concerns | 2 | 1.6% |
| Trust | 2 | 1.6% |
| Purchasing/Marketing | 2 | 1.6% |
| Parent Decisions | 1 | 0.8% |
| Performance Enhancers | 1 | 0.8% |
| Regulation | 1 | 0.8% |
| Social Welfare | 1 | 0.8% |
| Vaccinations | 1 | 0.8% |

**Table 3. Methods used in the extracted texts.**

| Study Characteristics | No. | % |
|---|---|---|
| *Study Type* | | |
| Qualitative | 14 | 11.2% |
| Quantitative | 102 | 82.3% |
| Mixed Qualitative and Quantitative | 8 | 6.5% |
| *Data Collection Method* | | |
| Survey | 70 | 56.5% |
| Choice Experiment | 9 | 6.5% |
| Qualitative Method | 8 | 6.5% |
| Document/Literature Analysis | 8 | 6.5% |
| Mixed Methods | 26 | 21.0% |
| Unspecified | 4 | 3.2% |
| *Likert or WTP Study* | | |
| Willingness-to-pay Study | 42 | 33.9% |
| Likert Scale Study | 49 | 39.5% |

experiment approach, three used different kinds of stated preference approach, and eight used econometric analyses without assessing consumer preferences. Other analysis methods were also used; 11 studies reported only descriptive statistics and univariate or bivariate analysis, and the final four studies reported only qualitative information. Of these 56 economics-focused studies, 25% primarily focused on antibiotics. The other studies investigated antimicrobial use as a component of animal rearing or a characteristic of food products themselves. Additionally, the challenge of antimicrobial resistance, with regard to public health, was a particular source of concern with only one study [18], which explored the environmental consequences of antimicrobial use and antimicrobial resistance development. Instead, antimicrobials were studied generally as a food safety issue, or with a set of other issues such as organic vs. conventional farming, animal welfare, and food quality. In most studies that utilized a willingness-to-pay model, people surveyed were willing to pay a premium for antibiotic-free products but this varied (between 0% and approximately 80%) depending on the geographic, social, and cultural settings investigated.

## Degree of consumer concern about antibiotics

Research on consumer perceptions of antibiotic use in animal agriculture encompasses a wide variety of subjects, and researchers utilized several measurement techniques, which challenges the ability to summarize findings among studies. Nevertheless, most studies found that consumer perceptions of antibiotic use exist along a spectrum. As described in the methods section, studies that gauged a level of concern about antibiotic use were coded as finding that consumers were "concerned about antibiotic use," "not concerned about antibiotic use," or had "mixed concern about antibiotic use." A total of 84.7% of studies were able to be classified in this way. The remaining studies measured other aspects of consumers perceptions, such as whether they know what an antibiotic-free label means [19, 20].

Among the literature investigated, 65.3% of studies concluded that consumers were concerned with antibiotic use in food production, 8.1% were not concerned, and 11.3% had mixed concern (see Table 4). Fig 2 summarizes the findings of studies that gauged consumer concern by tallying the number of studies by product type, method used, and level of concern. Likert scale surveys and willingness-to-pay studies dominate this research (73. 4%).

Table 4. Characteristics of studies that measured level of concern and reasons for concern.

| Consumer Concern Indicators | No. | % |
| --- | --- | --- |
| *Level of consumer concern for all 124 texts* | | |
| Concerned | 81 | 65.3% |
| Mixed Concern | 14 | 11.3% |
| Not Concerned | 10 | 8.1% |
| Study Did Not Measure Concern | 19 | 15.3% |
| *Reason for consumer concern from the 37 studies included in this analysis* | | |
| Safety | 9 | 24.3% |
| Human Health and Residues | 10 | 27.0% |
| Human Health and Resistance | 3 | 8.1% |
| Animal Welfare and Human Health | 1 | 2.7% |
| Animal Welfare, Human Health and Antimicrobial Resistance | 2 | 5.4% |
| Animal Welfare | 8 | 21.6% |
| Animal Welfare and Resistance | 1 | 2.7% |
| Production Practices | 2 | 5.4% |
| Social Responsibility | 1 | 2.7% |

| | Willingness to Pay | Likert | Qualitative | Total |
|---|---|---|---|---|
| **Beef** | Ø | | | Ø 1 |
| | ●●●● | ●●● | ● | ● 8 |
| | ◉ | ◉ | | ◉ 2 |
| **Pork** | | ●●●●●●● | | |
| | ●●●●●● | ● | ●● | ● 16 |
| | ◉ | ◉ | ◉ | ◉ 3 |
| **Poultry** | | | ØØ | Ø 2 |
| | ●●● | ●●● | ● | ● 7 |
| | | | | |
| **Dairy** | Ø | Ø | | Ø 2 |
| | ●●●● | ●●● | ●● | ● 9 |
| | | ◉ | | ◉ 1 |
| **Seafood** | | | | |
| | ●● | ● | | ● 3 |
| | | | | |
| **Multiple Products** | | Ø | | Ø 1 |
| | ●●● | ●●●●● | | ● 8 |
| | ◉◉ | ◉◉ | ◉ | ◉ 5 |
| **Generic "meat"** | Ø | ØØØ | | Ø 4 |
| | | ●●●●●●● ● | | ● 17 |
| | ●●●●●●● | ●● | | |
| | | ◉◉ | | ◉ 2 |
| **Total** | Ø 3 | Ø 5 | Ø2 | Ø 10 |
| | ● 29 | ● 32 | ● 7 | ● 68 |
| | ◉ 4 | ◉ 7 | ◉ 2 | ◉ 13 |

Ø        Study found no concern

●        Study found concern

◉        Study found mixed concern

**Fig 2. Tally of studies by food studied, methodology used, and level of concern about antibiotics that the study found.** The figure excludes studies that did not explicitly gauge a level of concern about antibiotics and studies that did not specify the product. Each dot is one study.

Consumers tended to demonstrate concern regardless of product type. The only exception was beef, a product in which consumer concern was slightly more mixed.

While the majority of studies (105 studies; 84.7%) found some measurable level of consumer concern about antibiotic use in food production, far fewer studies investigated their reasons. Among all studies, 29.8% (37 studies) investigated *why* consumers are concerned about antibiotics, and among this smaller subset of studies, personal health and safety comprise half of the reasons given (67.6% including the safety category and all categories with "human health" (Table 4)). The next most commonly cited reason was animal welfare, comprising 32.4% of such studies. It is notable that the evolutionary consequences of antibiotic use—the emergence of antibiotic resistant bacteria in the world—is mentioned in only four studies (10.8% of those that examined this reasoning, or 8.8% of the total number of studies) and this concern was always in combination with others. It is possible, however, that concerns about antibiotic resistance were an unmentioned or implied aspect of human health and safety concerns. Further specifying what consumers mean by "food safety" in this context is a possible avenue for further research.

Research about who is concerned about antibiotic use in food production is also relatively neglected in the literature; only 24% (n = 30) of included studies fitting this category. The most common descriptors across studies are gender (n = 13), income (n = 10), age (n = 9), and education (n = 6). In general, female, older, highly educated, and high-income were the demographic characteristics most consistently associated with consumer concern (Table 5). While

**Table 5. Summary of findings from studies that gauged the types of consumers concerned about antibiotic use.**

| Type of Characteristic | N[a] | Specific concern variables | "Not concerned" variables | Example Paper |
|---|---|---|---|---|
| Gender | 13 | female (10); males; both (situation dependent) | males (2) | Widmar 2017 |
| Age | 9 | over 65, over 70, older (4), younger, old/young (situation dependent) | young | Yuxiang 2019 |
| Income | 10 | higher income (8), lower income | higher income | Wolf et al. 2016 |
| Education | 6 | university degree, more educated (3) | more educated (2) | Steiner and Yang 2010 |
| Eating and Shopping habits | 4 | meat eaters, pork buying habits, shops at farmer's markets, household shopper | none | Bergstra et al. 2017 |
| Level of trust | 3 | high trust, low trust (2) | none | Muringai 2016 |
| Knowledge and Awareness | 3 | label readers, "health mavens", production knowledge | none | Smith et al. 2017 |
| Work | 3 | "housewives", union members, employed | none | Connor et al. 2008 |
| Political views | 3 | socially aware, conservatives social liberals | none | Bulut et al. 2021 |
| Ethical views | 3 | altruistic people, Individualizing moral foundation, believe that "organic" is better for cows | none | Lusk et al. 2007 |
| Religion | 3 | Protestants, Atheists, religiosity | none | Bergstra et al. 2017 |
| Race | 3 | non-white, Black, white | none | Steiner and Yang 2010 |
| Location | 2 | Montana, Quebec | none | Veeman and Lee 2007 |
| Family structure | 1 | parents with children under 6 | none | Tong 2011 |

[a]"N" is the total number of times the variable category was found to be significant across all papers. In sum, 52 variables across 30 different studies were found.

the findings for each of these features were consistent, there was at least one contradictory finding for each of these characteristics (e.g., one study found that men are more concerned about antibiotic use while all the others found more concern among women participants). Other personal identifiers included eating and shopping habits, level of trust, type of work, political views, ethical views, religion, race, awareness of the issue, location, and family structure. The results from these categories were found in few studies and without consistency across studies.

Although there are exceptions, questions about the politics of consumer choices and antibiotic use were largely unaddressed by these studies. One exception was Wolf et al. [21], which conducted a large survey that found two-thirds of consumers would vote to restrict antibiotic use to medical treatment only, and men were more likely to reject such a policy. Conversely, individuals with higher incomes and those exposed to animal welfare media were more likely to vote for such a policy. In another study, Goddard et al. [22] examined the link between people's moral foundations and their attitudes toward purchasing and voting decisions for various credence attributes. They found that those who agreed with individualizing moral foundation statements (i.e. having ethical concerns centered around impacts on individuals rather than having a commitment to the concerns of a wider social group) were more likely to purchase antibiotic-free products and also more likely to vote to ban such products compared to those who did not agree with such moral foundation statements. Finally, Lusk et al. [23] conducted a willingness-to-pay study that showed consumers were willing to pay more for antibiotic-free pork, and also were willing to pay a premium if there were a ban on such products.

## Discussion

Research that investigates consumer concern about antibiotic use in animal agriculture is gaining traction. This trend may relate to an increased public awareness and popularization of antibiotic-free and organic products, but longitudinal analysis was not conducted to confirm this theory.

Overall, consumer perceptions of antibiotic use in animal agriculture are distinctly negative. Among studies that measured the degree of consumer concern (n = 106), 77.4% found an appreciable level. Major threads of concern include consumer safety, health concerns around antibiotic residue on meat, the association of antibiotic use with poor animal welfare, and concern about antibiotic resistance. Concerned respondents are often wary of practices they associate with "contamination" [24]. Given the number of consumers who associated antibiotic use with poor animal welfare and food safety risk, it is possible that many misunderstand the role of antibiotics in animal agriculture. Such misperceptions, however, are still driving consumer views and behavior. While this review does not examine factors outside of antibiotic use, several studies found that genetically modified foods [24], pesticides, [25], and hormones [26] are also of concern to consumers.

Most studies indirectly measured antibiotic concern through credence labels (e.g., "raised without antibiotics" and "USDA Organic"), rearing practices, and food safety research in which antibiotic use was one of several related practices that were studied. Thus, in many cases, we had to extract the antibiotic-related findings from a study that was exploring a wider issue.

### Why are consumers concerned about antibiotics in animal agriculture?

While the reviewed literature demonstrated that consumers tend to be concerned about antibiotic use in animal agriculture, there are mixed findings as to why. Although few studies (24%) investigated their reasons, findings indicate interesting and inconsistent trends. Primarily, consumers are concerned about health and safety followed by concern that excessive antibiotic use is bad for animal welfare.

Some consumer reasons for concern indicate they may be ill-informed about animal agriculture production processes and antimicrobial uses. For example, some believed that administration of antimicrobials in animals may present health and safety hazards to consumers. Without further investigation, we cannot say what exactly those concerns are. One conjecture is that consumers believe that drug administration leads to antibiotic residues on or in animal products that could contribute to consumer exposure to active antimicrobial agents [27, 28]. However, the U.S. has strict regulations about antibiotic residues in animal products. For example, the USDA, in concert with the FDA and Environmental Protection Agency, founded the U.S. National Residue Program, which monitors residues in meat through its Compound Evaluation System. This ensures that the risk of exposure to antimicrobial residues in meat is low [29]. Similar regulatory efforts exist for non-meat animal products, such as milk. It is possible that consumers' concern for human health, in actuality, represents an unstated concern about antimicrobial resistance. However, none of the papers explored the potential conflation of these two terms. From a producer perspective, consumer concerns about animal welfare may appear similarly misguided. To that effect, some have argued that reducing on-farm antibiotic use is often worse for animal welfare because of the increased number of infections that tend to accompany the reduction [14, 30].

Consumers may not understand the nuances of antimicrobial use in animal husbandry, specifically in terms of disease treatment (i.e. treating clinically sick animals), metaphylaxis (i.e. administering antibiotics to a herd after animals are found to be sick), prophylaxis (i.e. administering antibiotics to a herd before animals become sick), and growth promotion/feed efficiency (i.e. antimicrobial administration to improve meat production). Consumer knowledge about these complexities is hard to evaluate, and no studies we could find addressed the terms with depth. Primarily, consumers associated antibiotic use with intensive animal production (such as CAFOs, factory farming) and lower animal welfare. The reality, however, is more nuanced, as animals may become infected with bacterial or other infectious agents even under optimized husbandry conditions and there is concern that organic practices could be harmful for animals if antibiotics are withheld when needed. According to many producers and veterinarians, maintaining good animal welfare means treating animals when they are sick, and practices that withhold antibiotics lead to worse animal welfare [30]. This view is more attuned to the complex trade-offs involved with using antibiotics. Such a view is one indication of the gulf in the attitudes between consumers and producers with regard to the relationship between antibiotic use and animal welfare, in addition to a difference in attitudes between organic and conventional producers. Singer et al.'s [30] survey shows that conventional (non-organic) producers are aware of this gap in understanding, even if consumers are not. They found that conventional producers felt consumers believe raising animals without antibiotics would have significant improvements for animal agriculture, contrary to these producers' views.

Compounded with the nuance of antimicrobial use in animal production are the complexities that exist between animals (i.e., cattle, chicken, turkeys, lamb) reared for consumption. The diversity of settings for animal production, specific species needs and threats, lifespan generalities concerning antibiotic in animal husbandry are difficult to establish. Some animals are raised in cages (i.e. chickens), others are raised outside in feed lots (i.e. cattle). Swine and chicken are typically raised in large defined housing systems, and some are raised on pasture. The method of animal husbandry affects their likelihood of contracting an illness and therefore the necessity for treatment and prevention [31]. Regardless of operation style, the bacterial flora composition of animals are dissimilar among species, including pathogens for both animals and humans. These differences, along with specific bans for use among animal agriculture species, are why some antimicrobial classes are used more in some species than in others. For example, according to the Animal User Fee Act data, in 2019, the cattle industry has purchased

81% of all cephalosporins used in animal agriculture, whereas the swine industry has purchased 85% of all lincosamides, and the turkey industry has purchased 66% of all penicillins [32].

These complexities of drug use and animal health across multiple species means that simplified labels can serve as an important signaling, though potentially misleading, device for consumers. This is suggested in Abrams et al.'s [33] qualitative study of pork consumers, where such labels become the key point of information for consumers who wish to avoid potential risks related to health and safety. While experts in animal production can point to statistics on the low prevalence of antibiotic residue found on meat, this work suggests that lay consumers tend to latch on to an easily understood, qualitative marker of risk such as a "raised without antibiotics" label. By attending only to the label, there is not deeper consideration of the alignment between this label, risk to human health and benefits or harms to animals.

Typically, when discordance is found between consumer perceptions and producer realities, it is often accompanied by a call to improve consumer education and address consumer knowledge gaps. We suggest that an education model designed around transferring expert knowledge about agriculture and antibiotics to consumers could be difficult to implement in terms of reaching consumers and garnering attention. More importantly, such a knowledge-deficit approach is likely to have limited efficacy in changing attitudes. This idea is supported by mounting evidence that such a model of science communication does not lead to the behavior or attitude change that is desired [34–36]. Instead, evidence from this review suggests that any model of educating a consumer should recognize the role of emotions and values, and directly address issues such as fear, trust, and uncertainty What is common across these studies, however, is that some consumers associate antibiotic use with a demonized view of the industrialized food system [30, 37]. Sonntag et al. [37], for example, found a wide range of consumer knowledge—from accurate understanding to misconception—but a fairly consistent attachment between antibiotic use and an industrial process that is regarded as unhealthy for chicken and, by extension, people.

"Better education" is not necessarily an inappropriate intervention, however, available evidence in this review suggests that knowledge is not the only factor that affects consumer perspectives, especially given the evidence that consumer antibiotic use concerns are tied to their negative feelings about modern industrial production systems [30, 37]. The relative paucity of research into why consumers are concerned about antibiotics shows that there is clearly more work to be done in this area. The literature to date has largely focused on how much consumers are willing to pay, or on quantifying the level of consumer concern. Unfortunately, the literature lacks extensive research on the emotive attachments that consumers have to food, the kinds of decision-making processes they make while in the grocery store, and the sorts of values beyond price they consider when making purchasing decisions (but with notable exceptions, see for example [23, 30, 38]. Researchers may do well to consider ethnographic or other qualitative techniques to address these questions.

## Who are the concerned consumers?

The literature has not comprehensively characterized individuals who may or may not be concerned about antibiotic use in animal agriculture. There were 24 studies that addressed this question, and of these studies, 14 different variables were identified as significant indicators of consumer concern. The most common significant variables were gender, age, education, and income. Collectively, these studies illustrate that older, highly educated, high-income females are most concerned about antibiotics use. Nevertheless, these findings were not consistent across studies, and other, less explored variables were implicated in these papers that paint a potentially more complex picture of the concerned consumer.

There were a host of other characteristics found to be of significance, but they were limited to just a few studies, with little consistency in findings. Individuals with both "high trust" and "low trust" in food safety were found to be concerned [39] along with "altruistic people" [23] and those with "individualizing moral foundations" [22]. In Connor et al.'s [40] study, "Protestants" and the "non-religious" were found to be concerned ("Catholic", the other religious choice in this study, was found to be a non-significant predictor). These differences could be the result of different methods and/or differences in study populations. Perhaps with more research more stable typologies will emerge as we have seen with gender, income, education, and age.

A few studies (n = 3) found that consumers with a higher level of knowledge and awareness about how antibiotics are used in agriculture tend to be concerned about antibiotics. Those with more knowledge seem to be more concerned, but as we discussed above, the kind of knowledge one has could greatly impact their stance on antibiotic use in animal industries. A high-knowledge consumer does not necessarily know specific information about antibiotic regimes and their role in animal production. Indeed, the components of antibiotic use that consumers were asked about in these studies were very basic, such as if they know what antibiotics are [41, 42]. Instead, "knowledge" can mean that a consumer understands the rules of thumb that labels provide, or has a general understanding of what antibiotics are, and how they are used in our food systems. We suggest here that there is a need for further research to understand the relationship between consumers' "antibiotic knowledge" and concern about antibiotics. For example, is a specific component of knowledge (or lack of) the reason for concern? And to what extent is the observed relationship confounded by income (which is typically higher among more educated consumers) and/or consumers' value or emotional attachments to food and animals, attitudes towards style of production (i.e. organic vs conventional) and their beliefs about safety and health? This kind of information can help to tailor a campaign to speak to values, motivation, and reasons to reassess their understanding of antibiotics in animal agriculture.

Finally, the relative dearth of explicitly policy and political affiliation studies is surprising given recent labeling changes and indicates a clear need for further research on the political orientations of consumers and approaches to relevant policy decisions. There is a growing visibility of consumption choices as a form of politics [43]. This can include campaigns to boycott particular products because of the product manufacturer's political views [44], or efforts to purchase products that meet ethical standards of production and trade [45, 46]. None of these political aspects of food consumption are covered by research into antibiotics and consumer preferences. Only three studies have linked political identity to views on antibiotics, but there is no consistency across studies, with both social liberals [42] and political conservatives [40] identified as expressing concern about antibiotics in agriculture. Numerous economics studies have established the degree to which consumers will, or will not, pay extra money for antibiotic-free products. But with few exceptions [23], none of these studies examine the extent to which these price preferences are related to political preferences with regard to agricultural policy. This is of particular concern because, as Paul et al. [47] note, a potential gap between the public's consumption and voting behavior can complicate supply chain decision-making due to "increased uncertainty regarding what 'social license' (e.g., freedom to operate) producers will maintain and what production practices will be accepted in the future" (pg. 102).

## Study limitations

There are several limitations to this review. First, this review should not be considered generalizable to populations outside of the U.S., Canada, the United Kingdom, and members-states of

the European Union. Secondly, we only included manuscripts written in English. This may have biased findings, given that Canada and the European Union have multiple official languages, and this review may have excluded relevant literature that was written in non-English languages. Similarly, selection bias may have occurred because we required that studies have primary data collection with transparent and extractable methods and results. Many excluded works were grey literature sources produced by industry members. Thus, this research is skewed to peer-reviewed literature conducted by academic institutions.

Finally, there are several important limitations regarding our variable that measured consumer concern across all of the studies. We developed this variable in an attempt to synthesize the broad, general findings about antibiotics in a way that is possible given data available in the reviewed studies and could be compared across all these 124 diverse studies. The studies analyzed in this paper use different methods, theoretical approaches, and modes of analysis so such a variable will necessarily miss some of the nuance between these studies. For willingness-to-pay studies, for example, it is possible that some respondents were concerned about antibiotics but did not place a higher price on antibiotic-free food because of resource constraints. These kinds of studies are more properly thought of as directly measuring "valuation" rather than concern, but we have interpreted a willingness-to-pay as the expression of a latent view of concern for the purposes of making a comparison across studies. The great majority of willingness-to-pay studies found that consumers were willing to pay more (82.1%), and only three found consumers unwilling to pay more. Since very few studies found that consumers were unwilling to pay, we believe our results are conservative in this regard. It is possible a slightly higher number of consumers are concerned, but lacked sufficient resources to express a higher willingness-to-pay. For Likert scale studies there are some difficulties in making comparisons because not all use the same scale. Scale sizes ranged from 3 to 7 points, with 50.3% of Likert studies using a 5-point scale. We used the percentage of respondents, or if appropriate, the average, above (or below) neutral as a way to categorize "concerned" versus "not concerned". More granular comparative reporting on the Likert studies was not possible (comparing the percentages at the most extreme values for example) because most (66.7%) did not report the disaggregated results of their study, opting instead to provide either aggregated scores, or averages. These limitations need to be considered when interpreting the synthesized results of this analysis.

The dominance of university researchers and U.S. studies likely resulted from inclusion criteria that required texts be in English and have primary data collection. We cannot say if a more expansive criteria would lead to others results. We also recognize that our criteria were limiting in the sense that non-academic types of literature (e.g. opinion pieces) were, with few exceptions, not captured and/or excluded. Future research into these other types of literature could be beneficial to further explain consumer perceptions and identify how these perceptions are acquired.

## Conclusion

This review was prompted by our interest in consumer perceptions about antimicrobial use in animal husbandry. Initial readings about this topic indicated that reasons for consumer concern are wide-ranging and consumers are often confused about the use of antimicrobials in animal agriculture. Despite their confusion, consumer perceptions are an important influence on animal agriculture practices. To understand what consumers see as the risks and benefits of antimicrobial use in animal agriculture, and to gauge which research and methodological gaps exist in this literature, we conducted a scoping review. Through an exhaustive search strategy and systematic screening process, we identified 124 texts that fulfilled our inclusion criteria.

We extracted relevant data from these texts for analysis, including the available data on consumer concern. The majority of studies used quantitative methods, willingness-to-pay studies and Likert surveys prominent among them, and were conducted by university researchers on U.S. populations. The studied products and themes varied.

Not every text measured consumer concern, and fewer assessed reasons for concern or identified characteristics of concerned people. Those that measured concern focused on antibiotic use, a priority to reduce antimicrobial resistance. The different topics of interest and methods used made synthesis of findings about consumer concern difficult. We developed a rubric to categorize each study's population into "concerned," "mixed concern," or "not concerned" regarding antibiotic use in animal agriculture. Most studies found some level of concern or mixed concern. Concern for human and animal welfare were the most common reasons cited. The animal welfare concern may derive from the consistent associations that consumers construe between antimicrobial use and industrial agriculture practices that they perceive as having negative consequences for the produced animals. It is notable that the emergence of resistant bacteria, which is a consequence of antibiotic use, is only mentioned in four studies and never as a study's explicit focus.

Our review reveals several methodological and conceptual gaps in the literature and point the way toward promising lines of research in the future. In terms of methodology, there is a paucity of qualitative studies. The majority of studies are either willingness- to-pay, Likert scale studies, or a combination of the two. Such quantitative studies can show consumer preferences and reveal trends across a population. Qualitative work involving interviews, focus groups and ethnographies can help flesh out the mechanisms for why consumers feel the way they do and even how they come to arrive at their opinions. The relative lack of qualitative work is also related to some empirical gaps found in the literature.

The persistence of the so-called "vote/buy" gap in the literature, where people will choose to ban a product that they will also purchase [47], suggests that people can take on differing identities and preferences depending on the situation, whether they are consumers in a grocery store or citizens in a voting booth. There is a relative paucity of studies that explore the relationship between one's political views of antibiotic regulation and the choices they make as a consumer. This research gap suggests a potentially fruitful line of research around antibiotics that more deeply interrogates the relationship between people's values toward food production, animals, and the environment and their attitudes toward the food they buy. Some studies in the review did do this (e.g. [37, 48]), but more work could be done.

This kind of work could help illuminate a second promising line of research, which is better understanding why consumers are concerned about antibiotic use in animal husbandry. Taken collectively, the results of this scoping review suggest that consumers have wide range of reasons for being concerned about antibiotic use, with little consistency across the range of studies that measured this. Health, safety and animal welfare were the most common reasons consumers gave, with only a few studies finding antimicrobial resistance as a stated reason for concern. It is unclear, however, exactly what consumers mean by "health and safety" and this term could, in fact, be expressing an unstated concern about antimicrobial resistance. Future work that more specifically interrogates the thought process behind consumer aversion to antibiotic use in animal agriculture could be promising.

## Supporting information

**S1 File. Bibliography for all extracted texts.** This contains citations for all 124 retained texts used for analysis.
(DOCX)

**S2 File. PRISMA scoping review checklist.** This is a checklist of all of the scoping review elements developed by PRISMA.
(DOCX)

## Acknowledgments

The authors sincerely thank Alison E. Stout and Genevieve Jones for their efforts in searching for grey literature and screening captured texts. This manuscript is the result of collaborations formed during the Human Dimensions of Antimicrobial Resistance in Agriculture Workshop in Nebraska City, NE, USA in May 2019.

## Author Contributions

**Conceptualization:** Jaime R. Barrett, Gabriel K. Innes, Guillaume Lhermie, Renata Ivanek, Amelia Greiner Safi, David Lansing.

**Data curation:** Jaime R. Barrett, Gabriel K. Innes, Kelly A. Johnson, David Lansing.

**Formal analysis:** Jaime R. Barrett, Gabriel K. Innes, Guillaume Lhermie, David Lansing.

**Investigation:** Jaime R. Barrett, Gabriel K. Innes, Kelly A. Johnson, Guillaume Lhermie, Renata Ivanek, Amelia Greiner Safi, David Lansing.

**Methodology:** Kelly A. Johnson.

**Project administration:** Kelly A. Johnson, David Lansing.

**Writing – original draft:** Jaime R. Barrett, Gabriel K. Innes, Kelly A. Johnson, Guillaume Lhermie, David Lansing.

**Writing – review & editing:** Jaime R. Barrett, Gabriel K. Innes, Kelly A. Johnson, Guillaume Lhermie, Renata Ivanek, Amelia Greiner Safi, David Lansing.

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
