## [Decision Letter · Decision Letter 0]

7 Sep 2021

PONE-D-21-21650Consumer perceptions of antimicrobial use in animal agriculture in the United States, Canada, and the European Union: A scoping reviewPLOS ONE

Dear Dr. Lansing,

Thank you for submitting your manuscript to PLOS ONE. After careful consideration, we feel that it has merit but does not fully meet PLOS ONE’s publication criteria as it currently stands. Therefore, we invite you to submit a revised version of the manuscript that addresses the points raised during the review process.

The main problem found in the manuscript is related to the some aspects of methodology, and redaction style. However, the manuscript could have been prepared more carefully, as some points remain incomprehensible, or the manuscript lacks some necessary information according to the comments of referees. Please review the referee comments and make your peer revision. Thanks for your hard work.

We look forward to receiving your revised manuscript.

Kind regards,

Arda Yildirim, Ph.D.

Academic Editor

PLOS ONE

Journal Requirements:

This research was funded by USDA-NIFA grant number 2019-67017-29114, which allowed for two in-person meeting of the co-authors, and USDA-NIFA grant number 2018-68003-27467, which provided research support for JRB and DL.

JRB, GKI, GL, RI, AGS, DL received funds from the United States Department of

Agriculture, National Institute for Food and Agriculture grant # 2019-67017-29114.

JRB, DL received funds from USDA-NIFA grant #2018-68003-27467

https://nifa.usda.gov/

The funders had no role in the study design, data collection and analysis, decision to

publish, or preparation of the manuscript.

Additional Editor Comments:

This MS deals with an interesting and important topic in animal husbandry. Nevertheless there are still some points of concern from the reviewers, before the manuscript can be accepted for publication. Please make MS title specific I suggest change “in animal agriculture” to “in animal husbandry”. The manuscript should be presented according to guidelines for authors of Plos One. I recommend major revision.

Reviewers' comments:

Reviewer's Responses to Questions

**Comments to the Author**

1. Is the manuscript technically sound, and do the data support the conclusions?

Reviewer #1: Partly

Reviewer #2: Yes

Reviewer #3: Yes

Reviewer #4: Yes

Reviewer #5: Yes

2. Has the statistical analysis been performed appropriately and rigorously? 

Reviewer #1: I Don't Know

Reviewer #2: N/A

Reviewer #3: N/A

Reviewer #4: N/A

Reviewer #5: N/A

3. Have the authors made all data underlying the findings in their manuscript fully available?

Reviewer #1: No

Reviewer #2: Yes

Reviewer #3: Yes

Reviewer #4: Yes

Reviewer #5: Yes

4. Is the manuscript presented in an intelligible fashion and written in standard English?

Reviewer #1: Yes

Reviewer #2: Yes

Reviewer #3: Yes

Reviewer #4: Yes

Reviewer #5: Yes

5. Review Comments to the Author

Reviewer #1: the manuscript conducts a scoping review on a topic of consumers perceptions towards antimicrobial use in animal agriculture

Comments

- the authors need to update the manuscript contribution

- the authors can suggest new research lines based on the review

- the review title suggests geographical focus. as a result the reader would expect a geographical classification description of main results and discussions in each target location.

-

Reviewer #2: The manuscript on 'Consumer perceptions of antimicrobial use in animal agriculture in the United States, Canada, and the European Union: A scoping review' is interesting and well-written. This can be published in PLOS ONE after minor revisions which is noted in the attached file.

Reviewer #3: The paper is timely and important. It is well written. Authors have done a scoping review on consumer perceptions about the antimicrobial use in animal agriculture in the US, Canada and European Union. I have one minor comment / suggestion:

'PRISMA guidelines about scoping reviews' may be explained briefly for general readers.

Reviewer #4: Major thoughts:

1. It is mentioned that the scope of the study was US, Canada and the EU. Were there articles published in English in these countries of interest but that were focused on investigating antibiotics perceptions in other countries (outside the EU, Canada and the U.S.? If so, it may be relevant to note this.

2. Comparisons between concern levels for antibiotics used between U.S., Canada and the EU could be interested to look into. Did the authors considered this?

3. An important finding highlighted by the authors is that “Antimicrobial resistance rarely registered as an explicit reason for concern.” While “human health” was a prominent reason for concern. I think that it is possible that consumers in the studies analyzed may have understood that “human health” included “antibiotics resistance”. This possibility should be considered and addressed.

Abstract:

The abstract says: “The most common and consistent features of these consumers were gender, age, income, and education.” But no findings from the analyzed articles were reported utilizing these demographics categories. I recommend removing this statement if it will not be directly used in the abstract.

The abstract says: “Overall, studies tended to be dominated by either willingness-to-pay studies or likert scale questionnaires (73.6% of all studies). The popularity of these methods may have contributed to the relative lack of studies that characterized worried consumer demographics or reasons for their perspectives.” But wtp and likert studies are not mutually exclusive with asking consumers why they hold certain views. This statement may misrepresent the extent of information that can be obtained from such study methodologies.

The abstract says: “We recommend more qualitative research into consumer views on this topic, which may better elucidate consumer decision-making and mentality” But it is not clear how asking qualitative questions to consumers will help with their decision making or mentality. It seems to me that educational campaigns may serve this purpose better then more qualitative research. Perhaps refrace?

Introduction:

122-124. “may influence” or “have some influence” may be more appropriate than “influence”

237. “We made this decision because these countries have similar regulatory environments and close trade associations” Do you mean similar regulatory environments when it comes to the use of antibiotics in animal agriculture? If so, you can include that in the manuscript

360. “The only exception was beef, a product in which consumer concern was mixed.” Do the authors have any idea or potential hypothesis as to why may this be the case?

364-371 “Among all studies, 29.6% (37 studies) investigated why consumers are concerned about antibiotics. Among these, personal health and safety comprise half of the reasons given (67.6% including the safety category and all categories with “human health”; see table 4). The next most commonly cited reason for concern was animal welfare, comprising 32.4% of studies where perspectives were evaluated. It is notable that the evolutionary consequences of antibiotic use—the emergence of antibiotic resistant bacteria in the world—is mentioned in only four studies (10.8% of those that examined reasoning) and this concern was always in combination with others.” Is it possible that consumers who participated in these studies referenced understood “personal health and safety” to encompass “evolutionary consequences of antibiotic use”? This is something that needs to be carefully addressed by understanding in detail the verbiage used in each study. This may change some of the main findings from this review.

Discussion:

479. “We reject that “better education” will lead to different results.” On what basis is this rejected?

493. “especially given that consumer antibiotic use concerns are tied to their negative feelings about modern industrial production systems” needs citation

519. “Both “Protestants” and “atheists” were also found to be concerned” Is this compared to other religious groups or faith associations?

523. “One small (three studies) but consistent finding is that a consumer with a high level of knowledge and awareness tends to be concerned about antibiotics” Is this “self-reported knowledge” if not, how was knowledge level measured?

535. “Consumers have different ways of evaluating agricultural production than producers, and the evidence so far suggests that is unlikely to change.” Please list this evidence

Conclusions:

600-602. “Similarly, more in-depth qualitative research is also needed on this topic because the overwhelming use of quantitative methods does not allow for a more nuanced understanding of consumer decisions.” Again, I am not sure is these two (quantitative methods research and qualitative questions) are mutually exclusive.

Reviewer #5: Dear authors:

Thank you for the opportunity to review your manuscript. Your manuscript addresses a pertinent topic of interest to many stakeholders. There are a number of issues with your characterization and conclusions that need to be addressed.

Major comments

I find several of the characterizations in your methodology problematic. In addition to some misnomers. For example, your categorization of different points of concern based on Likert scale ratings (page 11 L252-259) may be akin to comparing apples to oranges as the scales may have different ranges. I would advise considering the extreme ends and grouping the anything between the extremes together to avoiding the arbitrariness of using the average as the cut-off. Further, Willingness to pay (WTP) cannot be used as a measure of concern as WTP is strongly driven by income (ability to pay). There may be segments of consumers who care about antibiotic use but simply cannot afford to pay and therefore will report WTPs of zero. Instead of concern, why don’t categorized that as valuation?

Your conclusion that consumer perceptions of antibiotics use is overwhelmingly negative (L409) appear not to be supported by your results. Indeed, in the subsequent line you note that 77.4% found some level of concern. This according to your analysis is due to a misunderstanding of the what antibiotics use is (i.e. perceived contamination) (Brewer and Rojas 2008). It is also quite surprising that you appear to strongly oppose consumer education although your work severally indicates that consumers are misinformed about antibiotics, their effect and the intersection with animal welfare, human health and AMR. I see no basis in your work to reject the notion of increased consumer education about the issue.

The issue of antibiotics use in pork production is huge due to the nature of diseases and its impact on production. This is different from cattle, for example. At the very least, authors should discuss these issues. Yes, I agree your focus is very narrow, but a good background and context enriches your discussion. The current version of this manuscript is severally lacking a thorough discussion of the themes identified in the literature review.

Minor comments

Page 6 and 7 Line 143 and 144: You indicate that your study focuses on Canada, US and EU without providing a justification for the focus on these countries/regions. You wait until page 11 L237. The choice of the countries and the justification are an important piece of your study and have to be discussed much earlier in the text. Apart for the similarities in the regulatory environment. Are there other farm/consumer driven reasons for the choice?

L515- Trust in what? Is it institutional trust?

L523-High level of knowledge in what? Is it level of education or education specific to antibiotics?

Goddard (2019) is missing in the reference list.

6. PLOS authors have the option to publish the peer review history of their article (what does this mean?). If published, this will include your full peer review and any attached files.

Reviewer #1: No

Reviewer #2: **Yes: **Aurup Ratan Dhar

Reviewer #3: No

Reviewer #4: No

Reviewer #5: No

---

## [Author Response · Author response to Decision Letter 0]

22 Oct 2021

We would like to thank the reviewers for their careful reading of our manuscript. We believe the revised version is much better due to our engagement with the reviewers. For a detailed response to each reviewer and editor comment, please see our "Response to Reviewers" table at the end of the document. It is quite lengthy and in a table format that cannot be easily added in this field.

---

## [Decision Letter · Decision Letter 1]

23 Nov 2021

Consumer perceptions of antimicrobial use in animal husbandry: A scoping review

PONE-D-21-21650R1

Dear Dr. Lansing,

We’re pleased to inform you that your manuscript has been judged scientifically suitable for publication and will be formally accepted for publication once it meets all outstanding technical requirements.

Kind regards,

Arda Yildirim, Ph.D.

Academic Editor

PLOS ONE

https://www.researchgate.net/profile/Arda_Yildirim2

Additional Editor Comments (optional):

Thanks for your hard work.

Reviewers' comments:

Reviewer's Responses to Questions

**Comments to the Author**

1. If the authors have adequately addressed your comments raised in a previous round of review and you feel that this manuscript is now acceptable for publication, you may indicate that here to bypass the “Comments to the Author” section, enter your conflict of interest statement in the “Confidential to Editor” section, and submit your "Accept" recommendation.

Reviewer #1: All comments have been addressed

Reviewer #2: All comments have been addressed

Reviewer #3: All comments have been addressed

Reviewer #4: All comments have been addressed

2. Is the manuscript technically sound, and do the data support the conclusions?

Reviewer #1: Yes

Reviewer #2: Yes

Reviewer #3: Yes

Reviewer #4: (No Response)

3. Has the statistical analysis been performed appropriately and rigorously? 

Reviewer #1: Yes

Reviewer #2: N/A

Reviewer #3: N/A

Reviewer #4: (No Response)

4. Have the authors made all data underlying the findings in their manuscript fully available?

Reviewer #1: Yes

Reviewer #2: Yes

Reviewer #3: Yes

Reviewer #4: (No Response)

5. Is the manuscript presented in an intelligible fashion and written in standard English?

Reviewer #1: Yes

Reviewer #2: Yes

Reviewer #3: Yes

Reviewer #4: (No Response)

6. Review Comments to the Author

Reviewer #1: (No Response)

Reviewer #2: The authors have addressed all comments from the reviewers and it is well improved. I recommend to accept the manuscript for publication.

Reviewer #3: (No Response)

Reviewer #4: (No Response)

7. PLOS authors have the option to publish the peer review history of their article (what does this mean?). If published, this will include your full peer review and any attached files.

Reviewer #1: No

Reviewer #2: **Yes: **Aurup Ratan Dhar

Reviewer #3: No

Reviewer #4: No

---

## [Editor Report · Acceptance letter]

29 Nov 2021

PONE-D-21-21650R1 

Consumer perceptions of antimicrobial use in animal husbandry: A scoping review 

Dear Dr. Lansing:

I'm pleased to inform you that your manuscript has been deemed suitable for publication in PLOS ONE. Congratulations! Your manuscript is now with our production department. 

Kind regards, 

on behalf of

Prof. Dr. Arda Yildirim 

Academic Editor

PLOS ONE